# Online and Offline Disclosures of Unwanted Sexual Experiences: A Comparison of Reactions and Affect

**DOI:** 10.3390/bs15020102

**Published:** 2025-01-21

**Authors:** Melissa S. de Roos, Giorgia Caon, Elza Veldhuizen Ochodničanová

**Affiliations:** Erasmus School of Social and Behavioural Sciences, Erasmus University Rotterdam, 3062 PA Rotterdam, The Netherlands; caon@essb.eur.nl (G.C.); veldhuizenochodnicanova@essb.eur.nl (E.V.O.)

**Keywords:** sexual victimisation, unwanted sexual experiences, online disclosures, reactions to disclosures

## Abstract

People are increasingly turning to online settings to disclose very personal experiences, such as unwanted sexual encounters. Whilst the barriers to disclosure of such experiences and the positive effects of disclosure are well documented, little is known about the online disclosure experiences of survivors and victims of sexual violence, and no research has assessed differences between online and offline disclosures. This study assessed experiences of online and offline disclosures (*N* = 369; 86.4% female), focusing on people’s reasons for (non-)disclosure, the severity of people’s unwanted sexual experiences, the reactions they received to their disclosures, and how they felt about the disclosure. The results indicated differences between online and offline disclosures, with offline disclosures more strongly associated with negative responses than online disclosures. Moreover, people felt more positively about an online disclosure than they did about an offline disclosure. The interplay between these various factors and how people felt about their disclosure showed a different pattern across online and offline contexts. Implications and directions for future research are discussed.

## 1. Introduction

In recent years, there has been a sharp increase in awareness of, and conversations about, unwanted sexual experiences. Unwanted sexual experiences (USEs) refer to experiences of non-consensual sexual acts that can vary in type and severity and include a range of behaviours, such as sexual harassment, sexual coercion, and rape ([29]; [43]). Although most of these experiences remain undisclosed, some survivors and victims do choose to share their stories ([9]), and the frequency of such disclosures appears to be rising. For example, in the Netherlands, Victim Support saw a 25% increase in reports of sexual harassment and sexual violence in 2022 compared to 2021 ([46]). Similarly, the Crime Survey for England and Wales recorded a 32% increase in police reports of sexual offences in 2022 compared to the previous year ([16]). Multiple factors may influence a victim or survivor’s decision to disclose or not, such as the severity of the experience itself ([29]) or concerns regarding possible negative reactions to a disclosure ([9]). Additionally, global social media movements, such as the #MeToo movement, have impacted how sexual victimisation disclosures are made, with more people taking to online platforms to disclose their experiences ([22]).

The prevalence of sexual victimisation is alarmingly high. When observing childhood sexual abuse and adult sexual assaults together, studies report that 25% of women report suffering a USE, with one in five women experiencing rape ([3]). In research using student samples, it is reported that 20–50% of female university students, and upwards of 6% of male students, experience at least one USE throughout their time in university ([21]; [20]). Victims of childhood sexual abuse typically do not disclose their sexual abuse until they are adults ([36]), and although adult victims of sexual violence may be more likely to tell someone sooner, studies have found that most victims delay disclosure for a year or more ([52]). Of those who choose to disclose their experience, it is anticipated that most disclose to someone trusted, like a friend or family member, whereas far fewer disclose to a medical professional or police officer ([21]). Delaying a disclosure, or not disclosing in any capacity, has often been attributed to multiple so-called internal and external barriers ([3]). Many survivors and victims express they delayed disclosing their experience due to feelings of self-blame and shame ([3]; [21]). Furthermore, survivors and victims often delay disclosure to maintain privacy and avoid negative reactions ([9]; [21]).

Extensive research has investigated social reactions to disclosures of sexual victimisation, and their impacts on the victim or survivor disclosing. Although most people receive supportive reactions to their disclosures ([2]), the effects of a negative response to disclosure cannot be overstated. Negative responses are linked to a range of subsequent negative mental health consequences, such as anxiety and depression ([8]; [15]; [25]), whereas, unfortunately, positive responses only have a comparatively modest effect on adjustment and recovery following trauma ([11]; [42]; [52]). Moreover, receiving a negative response to a disclosure reduces the likelihood that the person will attempt another disclosure ([1]; [2]). Furthermore, studies demonstrate that when survivors experience victim-blaming after disclosure, the experience may feel like a “second assault”, which is a phenomenon known as “secondary victimisation” ([1]). Therefore, it remains vital, especially in the context of an ever-expanding societal discussion on inappropriate and unwanted sexual behaviour, that we improve our understanding of the effects of social reactions to such disclosures.

### 1.1. Disclosing Unwanted Sexual Experiences Online

The Internet provides an anonymous space for people to discuss and seek support for various experiences ([17]). Increasingly, young people especially are turning to the Internet for information on mental health ([31]), gender-related topics ([50]), sexuality ([27]), and sexual health ([19]). Online, people find a community and a safe space to discuss often very private issues ([34]; [38]).

In line with this, there has been a surge of online sexual victimisation disclosures in recent years. Viral social media movements, such as the #MeToo movement, which started in October 2017, and the #NotOkay movement of 2016, have made online disclosures of sexual victimisation increasingly prevalent ([23]). Some researchers suggest that online disclosures have become more common because disclosing using online platforms removes some of the barriers that victims and survivors face when deciding whether to disclose in real life ([22]). Real-life barriers faced by survivors and victims, such as shame and fear of negative repercussions, may be mitigated in online situations because anonymity can be protected ([47]). Furthermore, in some cases, online disclosures are seen as a way for victims and survivors to regain control over their experience and attribute blame to the perpetrator ([9]), to have their experiences validated ([54]), or to engage in activism and highlight the magnitude of the problem ([35]). Yet despite the positive intention of these movements to empower victims and survivors, they are not universally positive ([14]). Studies show that interacting with these movements has been linked with both positive and negative feelings ([14]; [37]). There is the possibility that individuals’ feelings and experiences within these social media movements could be impacted by the type of experience they disclose. Indeed, research suggests that mild and severe forms of sexual victimisation are more likely to be reported than those that fall somewhere in the middle of the spectrum ([29]). Since research in this area has exclusively focused on offline disclosures, it is unclear whether these findings translate to online disclosures.

Aside from the context within which a disclosure is made, the response to a given disclosure can significantly impact the individual’s experience of disclosing their USE. Importantly, negative responses are not exclusively tied to offline disclosures, and even though disclosures made online may bypass certain barriers that prevent disclosure, they are not immune to negative reactions from others ([44]). In fact, in the same way that anonymity in online settings may make survivors and victims feel more comfortable disclosing their experiences, it could encourage negative reactions from others, allowing them to act in ways they would not normally, which is referred to as the online disinhibition effect ([49]).

Nevertheless, as made evident by growing research, people are increasingly disclosing their unwanted sexual experiences in online spaces, but not much is known about how these disclosures aid or hinder their well-being. [22] ([22]) conducted a comprehensive systematic review summarising the research on online disclosures of sexual victimisation. They noted that responses to such disclosures tended to be positive and that negative responses were comparatively rare. Further, they found that people who disclosed online sometimes felt they had nowhere else to turn ([40]) or that it was a format used after an unpleasant offline disclosure experience ([41]). Participants reported various positive consequences of their online disclosures such as reduced self-blame ([24]), feeling a connection with other victims ([13]), and personal healing ([6]). However, negative consequences associated with public exposure as a victim were also reported ([13]). These findings suggest online disclosures can be a beneficial alternative to offline disclosures for victims and survivors. Nevertheless, as [22] ([22]) point out, the studies available for inclusion largely relied on qualitative methods and small samples. Further, only one study compared people who disclosed online with those who chose not to do so. As such, there is a need to directly compare online and offline disclosures using quantitative methods to determine whether any benefits are a product of disclosure per se, or whether online disclosures differ in important ways from offline disclosures.

### 1.2. The Present Study

The current study aims to extend current findings regarding disclosures of unwanted sexual experiences by investigating how online disclosures differ from offline disclosures. In doing so, we hope to improve our understanding of online disclosures and determine whether they are a beneficial method of disclosing USEs.

We examine differences in reasons for (non-)disclosure, reactions people received to their disclosure, how they feel about their disclosure, and the severity of their USEs. Our research questions are as follows: (1) Do people who disclosed online, offline, or both differ in terms of the severity of their USEs? (2) How do reactions to disclosures, reasons for (non-)disclosure, and severity of USEs relate to how people feel about their disclosure? (3) Do people who disclosed online, offline, or both differ in terms of their reasons for initial non-disclosure? (4) In their reasons for eventual disclosure, (5) do the reactions people receive to online vs. offline disclosures differ? And (6) do people feel differently about online disclosures compared to offline disclosures? Due to the exploratory nature of this research, we expected differences between online and offline experiences but not in any particular direction.

## 2. Methods

### 2.1. Participants

First, participants were recruited from various online support forums and social media pages focused on supporting victims of sexual violence. Each time, permission from a forum moderator was asked and granted before a link to the survey was posted. To supplement this sample, we linked it to the university student participant pool, where students could participate for participation credits. Participants were excluded if they had experienced suicidal ideation and/or had been hospitalised for suicidal ideation in the past six months, or if they reported no unwanted sexual experiences.

The survey was accessed by 520 participants who experienced at least one unwanted sexual experience as defined by the questionnaires used. Of this sample, the surveys of 124 participants were deleted because the participants completed less than half of the survey, a further 22 participant surveys were deleted because the participants finished the survey in less than five minutes, and two participant surveys were deleted because the participants were underage. This resulted in a sample of 369 participants, most of whom identified as female (86.4%), with a few identifying as male (9.2%) or non-binary (3.0%) and the remainder choosing to self-identify. The participants’ ages ranged from 18 to 69 (*M* = 23.42, *SD* = 7.66). Most participants identified as heterosexual (66.7%) or bisexual (22.2%). A majority identified as white (79.7%) and resided in Western Europe or the United States.

### 2.2. Materials

#### 2.2.1. Unwanted Sexual Experiences (SES-SFV; [30])

Unwanted sexual experiences after age 14 were assessed using the SES-LFV survey. The survey asks participants about seven experiences, ranging from fondling to anal and vaginal penetration with behaviour-focused phrasing. For each item, participants are asked to indicate whether this happened to them, and the method the perpetrator used (i.e., threatening to physically harm me or someone close to me). Based on responses, participants’ experiences are categorised as “sexual contact”, which denotes an affirmative answer to any of the fondling questions, “(attempted) coercion”, which covers verbal coercion strategies, or “(attempted) rape”, which involves (threats of) physical harm. Participants also report how often they experienced this (never, once, twice, or more than three times), and the sex of the perpetrator. Internal reliability was excellent in this sample (*α* = 0.95).

#### 2.2.2. Childhood Sexual Abuse Experiences Questionnaire ([7])

To assess unwanted sexual experiences before age 14, the CSAE Questionnaire was used. It asks participants whether, at any point before age 14, they experienced exhibitionism, sexual fondling or touching, or completed or attempted sexual intercourse. Participants are then asked if this happened once or multiple times, and their relationship to the perpetrator (i.e., stranger, family member, or acquaintance from school). Given the large number of questionnaires in this study, we chose to use this brief questionnaire to assess experiences in childhood rather than using the SES-SFV survey a second time to ask about experiences before the age of fourteen.

#### 2.2.3. Social Reactions Questionnaire ([51])

The participants were asked about the reaction to their disclosure (online and offline) through the SRQ. This 48-item questionnaire is made up of three general scales (turning against: “Pulled away from you”, unsupportive acknowledgement: “Wanted to seek revenge on the perpetrator”, and positive reaction: “Told you it was not your fault”) and seven specific scales (victim blame, treat differently/stigma, taking control, distraction, egocentric reactions, tangible aid, and emotional support). The participants reported how often they received each reaction on a 5-point Likert scale (1 = *Never*, 5 = *Always*). Internal reliability was good (Offline disclosures: *α* = 0.90; Online disclosures: *α* = 0.95).

#### 2.2.4. Positive and Negative Affect Scale ([53])

The participants indicated their feelings about their disclosure on the PANAS. This scale measures positive affect (e.g., Proud, Enthusiastic, Determined) and negative affect (e.g., Irritable, Ashamed, Afraid) using ten items per scale. The participants were asked to what extent their online and offline disclosure made them feel each emotion. Answers were scored on a 5-point Likert scale (1 = *Very slightly or not at all*, 5 = *Extremely*). Internal reliability was good (Offline disclosures: *α* = 0.89; Online disclosures: *α* = 0.88).

#### 2.2.5. Reasons for (Non-)Disclosure

To assess reasons for (non-)disclosure, we used two subscales from the Unwanted Sexual Experiences Scale developed by [28] ([28]). The first subscale assesses reasons for delaying or non-disclosure using seven items over subscales: shame (i.e., “I was embarrassed”) and ambivalence (“It was my fault as much as the other person’s”). Participants are asked to rate the importance of the reason for their decision to delay disclosure on a 4-point Likert scale (1 = “Not Important” to 4 = “Most Important”). Internal reliability in this sample was acceptable (*α* = 0.65). The next subscale assesses reasons for disclosure using thirteen items over four subscales: protecting others (“I was afraid someone else would get hurt if I didn’t tell”), weariness (“I told because I wanted it to stop so my life could go on”), internal reasons (“I told because I couldn’t hold it in any longer”), and environmental reasons (“Someone else convinced me to tell”). Scoring instructions and answer format are the same as the previous subscale. The internal reliability in this sample was good (*α* = 0.75).

### 2.3. Procedure

The participants accessed the survey online through an anonymous link. After providing consent, they first filled out the SES-SFV survey and the CSAE Questionnaire. If they reported no experiences, they were thanked for their participation and linked to the end of the survey. If they did report experiences, they were asked if they had ever disclosed online, including through reposting the #MeToo hashtag. If they answered affirmatively, they filled out the SRQ and PANAS about their online disclosure experience. Following this, they were asked if they disclosed this to someone in person. Again, if they answered “yes”, they were asked to fill out the SRQ and PANAS about their offline disclosure experience. They were also asked who they first disclosed to. Finally, demographic information was collected. At the end of the survey, the participants were debriefed and presented with a list of international and national support organisations they could reach out to if they experienced any negative consequences as a result of their participation in this study.

## 3. Results

We first computed descriptive statistics of reasons for (non-)disclosure, reactions people received to their disclosure, how they felt about their disclosure, and the severity of their unwanted sexual experience, irrespective of what type of disclosure they made. Where people made both an online and offline disclosure, their scores across online and offline experiences (reactions and affect) were averaged. The results are displayed in Table 1.

One participant reported unwanted experiences only in childhood. A further 45.5% reported experiences after the age of fourteen only, and 54.2% of participants reported experiences both in childhood and after the age of fourteen. Of the participants who reported experiences after the age of fourteen, most were victimised by a male perpetrator only (86.7%). The SES also asks participants if they think they have ever been raped, and 27.2% responded affirmatively, even though 49.6% of the sample endorsed the behaviour-focused item asking about rape. As mentioned, 17.3% of the sample never disclosed and thus were excluded from analyses looking at disclosure experiences. Of the full sample, 35.3% told someone about their experience within the same week, and 25.5% waited longer than a year to tell someone. The most likely recipients of disclosure were a friend (64.0%) or a parent (10.5%). Formal disclosures were rare; a total of 2.6% of participants first disclosed to a healthcare professional. And 0.6% first disclosed to police.

Our first research question concerned differences in the severity of USEs across the different disclosure groups. The results are detailed in Table 2. The absolute percentages represent the proportion of participants who had had such an experience, whereas the mutually exclusive category indicates the most severe category participants fell into. For each disclosure group, most participants’ reported experiences were of the most severe category, namely rape.

To answer our second research question, how reactions to disclosures, reasons for (non-)disclosure, and severity of USEs relate to how people feel about their disclosure, we examined bivariate relationships for the three groups who disclosed (online only, offline only, or both). The results are displayed in Table 3. Reactions to disclosures were most strongly related to negative affect and reactions were more strongly related to affect for offline rather than online disclosures. The severity of sexual victimisation was associated with a stronger affect in general, but a negative affect in particular. People who made both online and offline disclosures showed the strongest effect of severity on negative affect. Reasons for non-disclosure were more strongly related to negative affect and to affect following offline disclosures, but this was more varied for reasons for disclosure.

Our third research question concerned the differences between the disclosure groups in their reasons for non-disclosure. We compared reasons for non-disclosure between the four groups (including non-disclosers) using a MANOVA with shame and ambivalence as dependent variables and the disclosure group as independent variables. The multivariate test was significant (Pillai’s Trace = 0.06, *F* (6, 578) = 2.92, *p* = 0.008). Examination of between-subject effects indicated a significant effect was present for shame (*F* (3, 289) = 5.70, *p* < 0.001) but not ambivalence. Post hoc analyses with Bonferroni correction showed the people who made online and offline disclosures reported shame as a stronger reason for non-disclosure (*M* = 2.56, *SE* = 0.10) than people who never disclosed (*M* = 2.08, *SE* = 0.10; *p* = 0.004) or people who only disclosed offline (*M* = 2.14, *SE* = 0.06; *p* < 0.001).

To answer our fourth research question, which focused on group differences in reasons for disclosure, we first excluded the people who never disclosed. To analyse reasons for disclosure, we again conducted a MANOVA with the disclosure group as an independent variable, and the four reasons for disclosure (protect others, weariness, internal reasons, and environmental reasons) as dependent variables. The multivariate test was significant (*F* (8, 568) = 3.73, *p* < 0.001), as were the between-subject tests for all four reasons of non-disclosure (protect others: *F* (2, 289) = 8.09, *p* < 0.001; weariness: *F* (2, 289) = 9.80, *p* < 0.001; internal reasons: *F* (2, 289) = 3.30, *p* = 0.038; and environmental reasons: *F* (2, 289) = 4.42, *p* = 0.013). For all four reasons for disclosure, post hoc analyses with Bonferroni correction indicated the significant differences were between the group who disclosed both online and offline and the group who only disclosed offline (see Table 4).

Our fifth research question looked at the difference in reactions people received online and offline, and our sixth research question concerned how people felt about their disclosures. To answer these questions, we performed separate analyses as follows. First, we conducted a between-subject analysis, including only participants who had made either an online or an offline disclosure. Next, we analysed participants who had made both disclosures separately in a within-subject design. For the between-subject analysis comparing online and offline disclosure experiences, we performed two MANOVAs, one looking at the reactions (positive, unsupportive attacking, and turning against) they received and the other looking at how they felt about the disclosure (positive and negative affect). No significant differences were found in the reactions they received. The MANOVA looking at the affect following disclosure was significant (*F* (2, 227) *=* 3.17, *p* = 0.044). Between-subject tests showed the difference was in positive affect (*F* (1, 229) = 5.63, *p* = 0.019), with people who made online disclosures reporting significantly greater positive affect *(M* = 2.25, *SD* = 0.14) than those who made offline disclosures *(M* = 1.91, *SD* = 0.05). We then performed a series of paired sample *t*-tests to compare the online and offline experiences of participants who had engaged in both. The results are displayed in Table 5. Offline disclosures were associated with greater negative responses than online disclosures, greater positive affect, and lower negative affect.

## 4. Discussion

In light of the recent increase in USE disclosures made in online contexts, the aim of this study was to examine people’s experiences of online and offline disclosures of unwanted sexual experiences. Specifically, we were interested in differences in reasons for (non-)disclosure, reactions people received to their disclosures, and how they felt about their disclosures. We also looked at the severity of USEs.

In terms of disclosure, 17.3% of our sample never disclosed to anyone and thus were excluded from analyses looking at disclosure experiences. About a third of the participants told someone within the first week of the USE, and about one in four waited longer than a year to first tell someone. Consistent with the literature, most disclosures were made to a friend. Interestingly, only one participant reported a USE in childhood only; the remainder of our sample reported experiences only after age fourteen or experiences in childhood and adulthood, which lends support to the notion that people abused in childhood may be at increased risk of being revictimised later in life ([26]; [45]).

Looking at the severity of USEs in our sample, we found that when focusing on the most severe reported experience for mutually exclusive categories, for most people across all disclosure groups, their most severe experience was rape. Given the high absolute prevalence of the other categories, this suggests that for a substantial proportion of people, their experiences escalate from less severe (sexual contact) to the most severe, which is consistent with grooming practices employed by perpetrators, particularly in cases of child sexual abuse (e.g., [56]). However, it also hints at heightened vulnerability consistent with theories seeking to explain the risk of revictimisation ([4]). Notably, we found a difference in people’s answers to a behaviourally focused item asking about rape, which was endorsed by 49.6% of the total sample, but when asking about “rape”, only 27.2% responded affirmatively. This discrepancy indicates a common finding that people may not recognise their experiences as rape, even though their experiences meet a legal threshold of what constitutes rape (e.g., [55]). The literature on the potential effects of such “unacknowledged rape” is mixed, as it has been associated with maintaining a relationship with the perpetrator and increasing the risk of revictimisation, but it may also protect the individual from negative consequences, at least in the short-term, through minimising their experience ([32]).

Consistent with the literature, the reactions people received to their disclosure were most strongly related to negative affect rather than positive affect ([11]; [42]; [52]). Further, reactions to disclosures were more strongly related to the overall affect of offline disclosures compared with online disclosures. This stronger association may be due to the more personal nature of an offline disclosure, which may be more likely to be made to a trusted individual. Any reaction from a close relationship is likely to lead to a stronger emotional reaction than a response from a more anonymous person online ([5]). Similarly, an offline disclosure may be more immediate, where people cannot hide their initial reactions as well as they could in an online setting. Further, the effect of the severity of USEs on negative affect following disclosure was particularly salient for those who made both online and offline disclosures. This finding hints at a possibility that individual differences exist between those who choose to disclose online and those who do not, both in terms of the individual and the details of their USE. As [22] ([22]) pointed out, very little research has paid attention to the characteristics of the discloser or of the USE and how these affect the experience of an online disclosure. Future research should include such characteristics to improve our understanding of the individual–situation interaction.

We then zoomed in on the differences between online and offline disclosures across various variables. With regards to reasons for non-disclosure, we found that shame was a significantly stronger reason for initial non-disclosure for people who made both an online and offline disclosure compared with people who never disclosed or only disclosed online. This is interesting as it suggests shame is a very strong barrier to initial delayed disclosure in this group, which could be an explanation for the finding of [40] ([40]), that people turned to an online context to disclose because felt they had no other options. Further, these people may feel more at ease making an online disclosure first, which can be relatively anonymous and safe in the sense that the person is not confronted with others’ immediate reactions. Indeed, research on the disclosure of various types of personal information suggests that online disclosures may boost self-confidence and self-efficacy, which could give the individual the courage needed to make an offline disclosure (e.g., [5]; [12]). Unfortunately, we did not ask people about the order of their disclosures (i.e., did they first disclose online and then offline), which is something future research should include.

We found similar results for reasons for disclosure, in that the group of people who made both an online and an offline disclosure reported that all four reasons for eventual disclosure were significantly more important compared with people who only disclosed online. Whilst we did not specifically assess this, previous research found reasons for disclosure that may be unique or more salient for online disclosures than offline disclosures, although they focused on online disclosures without making a comparison between the two. A significant factor in online disclosures is other-oriented disclosure ([23]), which can occur in offline settings, but has a much broader potential range in online settings. For example, filling a support gap for other victims has been cited as a reason for online disclosure ([24]), and a broader motivation to support other victims (e.g., [33]). Future research should focus on such reasons that have been identified as motivators for online disclosures and compare how they differ from motivations to disclose offline.

When comparing the responses people received to their disclosures, we found no differences between online and offline disclosures for people who made either an online or an offline disclosure. However, when comparing the experiences of people who made both an online and an offline disclosure, offline disclosures garnered more negative reactions, whereas there was no significant effect for positive reactions. These findings suggest that when people can compare their experiences of online and offline disclosures directly, the online disclosure results in fewer negative reactions, which is consistent with research indicating that online reactions tend to be positive and negative reactions are rare ([23]).

Finally, we looked at how people felt about their disclosures. Comparing online and offline experiences of people who had only made one type of disclosure (online or offline), people who disclosed online reported significantly greater positive affect than those who disclosed offline. This pattern was also reflected in comparisons between the experiences of people who made both an online and an offline disclosure; they reported greater positive affect regarding their online than their offline disclosure. For negative affect, no effect of type of disclosure was found for people who only made one type of disclosure, but for those who made both types of disclosures, they felt more negatively about their offline experiences than they did about their online experiences. These findings provide partial support for the idea that online experiences can result both in more positive and less negative reactions. This may be due to several reasons. Victims and survivors may monitor the online spaces and see how other people’s disclosures are received by the online community before deciding to make their own disclosures. In an offline context, such observations are likely to be limited, but we do know that victims and survivors are sensitive to feedback received to a first disclosure, which impacts the likelihood of subsequent disclosures ([48]). By monitoring online spaces, they are able to observe feedback others receive to their disclosure and thus make a prediction about how their own disclosure experience may unfold. Future research should focus on different online platforms to examine which online formats lend themselves particularly well to maximising the positive effects of online disclosure.

Similarly, people who disclosed online reported more positive affect about their experience than those who disclosed offline. Separate from the nature of reactions received, it could be the case that an online disclosure garners a greater amount of support as the potential audience is larger than in an offline disclosure. Further, the potential of online disclosures to contribute to activism ([18]) and challenging stigma associated with sexual victimisation ([6]) may facilitate a strong, positive experience that enhances the person’s ownership of their experience and may give them a sense of community and contributing to a greater cause. The findings also partially support the notion that online disclosures result in lower negative affect than offline disclosures, suggesting they can be a valid and beneficial form of disclosure.

### 4.1. Implications and Suggestions for Further Research

The findings from this study offer key insights into the various experiences relating to online and offline disclosures of USEs, with implications for both theoretical understanding and practical application. Firstly, the distinct emotional and social dynamics observed in the findings underscore the critical need for support services to adapt their approaches based on the differing dynamics of online and offline disclosures. For example, offline disclosure interventions should prioritise reducing negative reactions from recipients, as these have been shown to correlate strongly with heightened negative affect among survivors ([25]). Training programmes for family or friends could therefore be initiated to improve understanding of trauma responses and effective communication. Conversely, online platforms offer a unique advantage by allowing survivors to disclose anonymously, often reducing the risk of direct negative interpersonal feedback ([23]). Thus, services should invest in building and promoting online resources tailored to the emotional safety and accessibility these platforms provide. These measures would align with broader research suggesting that tailored online interventions are effective in addressing digital forms of trauma disclosure ([22]).

Furthermore, this study’s findings relating to the prevalence of negative reactions to offline disclosures emphasise the urgent need for targeted public education initiatives. Educational campaigns should aim to cultivate empathy and promote supportive responses among potential disclosure recipients, focusing on the detrimental effects of unsupportive reactions. Previous research has shown that positive social support mitigates the psychological impact of trauma ([10]), suggesting that fostering a culture of empathetic listening could significantly enhance survivors’ recovery processes.

Considering this study’s limited temporal scope, future research should adopt a longitudinal approach to examine the long-term outcomes of disclosures made in online versus offline contexts, focusing on how emotional and social dynamics evolve over time. Tracking survivors’ well-being across different timelines and contexts would provide deeper insights into the sustained impact of disclosure settings on mental health and recovery. Additionally, further investigation into how intersectional factors—such as gender, age, cultural background, and socio-economic status—influence the choice and experiences of disclosure across settings is necessary. Prior studies have focused on disparities between Black and white survivors ([25]); however, understanding these complexities could inform culturally sensitive and demographic-specific interventions, ensuring that support systems are tailored to meet diverse needs effectively.

### 4.2. Limitations

This study is the first, to the best of our knowledge, to compare people’s experiences of online and offline disclosures of sexual victimisation, both between subjects and within subjects. Nevertheless, some limitations can be identified. First, the group of participants who only made an online disclosure was very small, which means most of the significant differences were found between people who disclosed offline only and those who made online and offline disclosures. Future research should focus on specifically recruiting people who only made online disclosures, as it is likely their motivations and experiences will differ from those who made online and offline disclosures. Second, our sample consisted primarily of young, heterosexual, white women. Although young white women seem to be the demographic most likely to engage in online disclosures of sexual victimisation ([39]), it is essential to include more diverse groups of victims and survivors who may face unique barriers to offline disclosure due to their gender or ethnicity.

## 5. Conclusions

This study assessed differences in online and offline disclosure experiences of unwanted sexual experiences. Through a quantitative survey method, we found that people who made online and offline disclosures reported stronger reasons for disclosing. Many of the differences we found were between those who made both types of disclosures and those who only made offline disclosures. Overall, online disclosures were associated with fewer negative responses and more positive affect regarding the disclosure experiences. More research is needed to examine the characteristics of disclosers, disclosure content, as well as the long-term outcome of such disclosures to determine under which circumstances the positive effects of an online disclosure are maximised, whilst negative effects such as increased exposure and public scrutiny are minimised.

## Figures and Tables

**Table 1 behavsci-15-00102-t001:** Descriptive statistics of all variables.

	*N*	*M* (SD)	Range	Skew	Kurtosis
**Reasons for non-disclosure**					
Shame	296	2.23 (0.73)	1–4	0.34	−0.71
Ambivalence	292	1.61 (0.69)	1–4	1.08	0.55
**Reasons for disclosure**					
Protect others	301	1.53 (0.59)	1–4	1.18	1.08
Weariness	299	1.88 (0.78)	1–4	0.69	−0.29
Environmental reasons	300	1.17 (0.35)	1–4	3.10	15.14
**Disclosure reactions**					
Turning against	290	0.51 (0.63)	0–3.77	1.91	4.20
Unsupportive attacking	290	0.74 (0.62)	0–2.80	1.12	0.88
Positive reaction	292	1.94 (0.79)	0–3.63	−0.36	−0.40
**Disclosure affect**					
Positive	289	2.02 (0.71)	1–4.25	0.64	−0.25
Negative	290	2.62 (0.82)	1–4.60	0.14	−0.80
**USE severity**	301	4.82 (1.50)	2–6	−0.94	−0.64

**Table 2 behavsci-15-00102-t002:** Severity of unwanted sexual experiences by disclosure group.

	Non-Disclosure (*n* = 59)		Online Disclosure (*n* = 29)		Offline Disclosure (*n* = 218)		Online and Offline Disclosure (*n* = 65)	
	Absolute	Mutually Exclusive	Absolute	Mutually Exclusive	Absolute	Mutually Exclusive	Absolute	Mutually Exclusive
Sexual contact	86.4	13.6	100	20.7	93.1	16.1	96.9	10.8
Attempted coercion	50.8	13.6	58.6	10.3	53.2	6.0	67.7	1.5
Coercion	42.2	16.9	58.6	6.9	43.6	8.3	67.7	18.5
Attempted rape	47.5	13.6	51.7	17.2	53.2	16.5	61.5	16.9
Rape	40.7	40.7	44.8	44.8	51.8	51.8	52.3	52.3
Discrepancy	3.4%		6.9%		2.8%		12.3%	

*Note:* Discrepancy refers to the number of people who answered the behaviour-focused question affirmatively, but did not endorse the item explicitly asking about “rape”.

**Table 3 behavsci-15-00102-t003:** Correlations between victimisation experiences, reasons for (non-)disclosure, reactions to disclosure, and positive and negative affects following disclosure.

	Positive Affect				Negative Affect			
	Online	Offline	Both		Online	Offline	Both	
			Online	Offline			Online	Offline
Severity of USE	0.01	0.18 *	0.34 **	0.21	−0.08	0.21 **	0.39 **	0.42 ***
**Reactions**								
Turning against	−0.17	0.09	0.20	0.07	0.10	0.48 ***	0.35 **	0.28 *
Unsupporting Attacking	−0.07	0.21 **	0.22 ^†^	0.17	0.34 ^†^	0.54 ***	0.47 ***	0.40 ***
Positive	0.35 ^†^	0.33 ***	0.36 **	0.44 ***	0.34 ^†^	0.26 ***	0.40 **	0.22 ^†^
**Non-disclosure**								
Shame	0.33	0.16 *	0.30 *	0.18	0.10	0.61 ***	0.37 **	0.47 ***
Ambivalence	0.14	0.10	<0.01	0.03	0.23	0.02	−0.13	0.04
**Disclosure**								
Protect others	0.06	0.26 ***	0.19	0.21	0.16	0.47 ***	0.35	0.45 ***
Weariness	0.53 **	0.23 ***	0.18	0.25 ^†^	0.32	0.31 ***	−0.04	0.13
Internal reasons	0.27	0.18 **	0.32 *	0.05	0.37 ^†^	0.37 ***	0.04	0.23 ^†^
Environmental reasons	−0.29	0.07	0.22 ^†^	0.01	0.27	0.19 **	0.23 ^†^	0.21

*** *p* < 0.001, ** *p* < 0.01, * *p* < 0.05, and ^†^
*p* < 0.1.

**Table 4 behavsci-15-00102-t004:** Multiple comparison analyses of reasons for disclosure between offline and online and offline disclosure groups.

	Online and Offline		Offline Only		*p*	95% CI
	Mean	SE	Mean	SE		
Protect others	1.78	0.07	1.45	0.04	<0.001	0.13; 0.53
Weariness	2.24	0.10	1.77	0.05	<0.001	−0.74; −0.20
Internal reasons	2.67	0.12	1.13	0.02	0.03	0.02; 0.66
Environmental reasons	1.25	0.40	1.13	0.02	0.014	0.02; 0.24

**Table 5 behavsci-15-00102-t005:** Paired sample *t*-tests of reactions to online and offline disclosures for participants who made both disclosures.

	Online Disclosure	Offline Disclosure	*t*	*p*
Turning against	0.67 (0.71)	0.96 (0.95)	−2.77	0.008
Unsupporting attacking	0.83 (0.69)	1.15 (0.80)	−3.72	<0.001
Positive reaction	1.87 (0.96)	1.97 (0.77)	−0.79	0.435
Positive affect	2.34 (0.80)	2.14 (0.76)	2.41	0.019
Negative affect	2.45 (0.73)	2.84 (0.83)	−4.30	<0.001

## Data Availability

Data are contained within the article.

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
