# Peer review of "Online and Offline Disclosures of Unwanted Sexual Experiences: A Comparison of Reactions and Affect"

_behavsci, 2025, doi:10.3390/bs15020102_

Round 1
Reviewer 1 Report
Comments and Suggestions for Authors
Thank you for your submission. I enjoyed reading your manuscript on this important topic. I have two minor comments:
1) It would be useful if a statement is added to the methods section that explicitly outlines how the severity of USEs was measured
2) Please improve the formatting of the tables (particularly table 3)
Author Response
We thank the reviewer for their comments and the time taken to review our manuscript.
Comment 1: It would be useful if a statement is added to the methods section that explicitly outlines how the severity of USEs was measured.
Reply: We agree with the reviewer's comment. On page 4, lines 171-174, we have added the following information: Based on responses, participants’ experiences are categorised as “sexual contact”, which denotes an affirmative answer to any of the fondling questions, “(attempted) coercion”, which covers verbal coercion strategies, or “(attempted) rape”, which involves (threats of) physical harm.
Comment 2: Please improve the formatting of the tables (particularly table 3)
Reply: We thank the reviewer for their comment. To comply with the journal's typesetting and margins, the only way to make tables 2 & 3 more reader-friendly was to decrease the font size, which we have done for both these tables, on pages 6 & 7 respectively.
Reviewer 2 Report
Comments and Suggestions for Authors
Thank you for opening another dimension of the Internet and the possible value of social media.

Author Response
We thank the reviewer for their comments and the time taken to review our manuscript.
Reviewer 3 Report
Comments and Suggestions for Authors
The introduction effectively analyzes the phenomenon and supports its discussion with accurate bibliographical references. The focus on online disclosure experiences of survivors and victims of sexual violence is both highly relevant and thought-provoking. This topic addresses a significant gap in the literature by comparing online and offline disclosures. The methodology employed is well-suited to the research objectives. Additionally, the discussions, limitations, and conclusions are articulated clearly and comprehensively.
Author Response
We thank the reviewer for their thoughtful comments and the time taken to review our manuscript.